# The Predicted Splicing Variant c.11+5G>A in *RPE65* Leads to a Reduction in mRNA Expression in a Cell-Specific Manner

**DOI:** 10.3390/cells11223640

**Published:** 2022-11-17

**Authors:** Irene Vázquez-Domínguez, Lonneke Duijkers, Zeinab Fadaie, Eef C. W. Alaerds, Merel A. Post, Edwin M. van Oosten, Luke O’Gorman, Michael Kwint, Louet Koolen, Anita D. M. Hoogendoorn, Hester Y. Kroes, Christian Gilissen, Frans P. M. Cremers, Rob W. J. Collin, Susanne Roosing, Alejandro Garanto

**Affiliations:** 1Department of Human Genetics, Radboud University Medical Center, 6525 GA Nijmegen, The Netherlands; 2Donders Institute for Brain, Cognition and Behaviour, Radboud University Medical Center, 6525 GD Nijmegen, The Netherlands; 3Department of Neurology, Donders Institute for Brain, Cognition and Behaviour, Radboud University Medical Center, 6525 GA Nijmegen, The Netherlands; 4Department of Pediatrics, Amalia Children’s Hospital, Radboud University Medical Center, 6525 GA Nijmegen, The Netherlands; 5Radboud Institute of Molecular Life Sciences (RIMLS), Radboud University Medical Center, 6525 GA Nijmegen, The Netherlands; 6Department of Ophthalmology, Radboud University Medical Center, 6525 GA Nijmegen, The Netherlands; 7Division Laboratories, Pharmacy and Biomedical Genetics, Clinical Genetics, University Medical Center of Utrecht, 3584 CX Utrecht, The Netherlands

**Keywords:** inherited retinal diseases, Leber congenital amaurosis, *RPE65* gene, retinal pigment epithelium (RPE), induced pluripotent stem cells (iPSCs), cell-specific defects, iPSC-derived models

## Abstract

Pathogenic variants in *RPE65* lead to retinal diseases, causing a vision impairment. In this work, we investigated the pathomechanism behind the frequent *RPE65* variant, c.11+5G>A. Previous in silico predictions classified this change as a splice variant. Our prediction using novel software’s suggested a 124-nt exon elongation containing a premature stop codon. This elongation was validated using midigenes-based approaches. Similar results were observed in patient-derived induced pluripotent stem cells (iPSC) and photoreceptor precursor cells. However, the splicing defect in all cases was detected at low levels and thereby does not fully explain the recessive condition of the resulting disease. Long-read sequencing discarded other rearrangements or variants that could explain the diseases. Subsequently, a more relevant model was employed: iPSC-derived retinal pigment epithelium (RPE) cells. In patient-derived iPSC-RPE cells, the expression of *RPE65* was strongly reduced even after inhibiting a nonsense-mediated decay, contradicting the predicted splicing defect. Additional experiments demonstrated a cell-specific gene expression reduction due to the presence of the c.11+5G>A variant. This decrease also leads to the lack of the RPE65 protein, and differences in size and pigmentation between the patient and control iPSC-RPE. Altogether, our data suggest that the c.11+5G>A variant causes a cell-specific defect in the expression of *RPE65* rather than the anticipated splicing defect which was predicted in silico.

## 1. Introduction

Inherited retinal diseases (IRDs) are a highly heterogenous group of neurodegenerative disorders leading to a visual impairment as a result of the progressive loss of photoreceptor and/or retinal pigment epithelium (RPE) cells [1]. IRDs affect 1 in 3000 people worldwide [2], and causative variants in more than 270 genes have been reported (https://sph.uth.edu/retnet/ accessed on 30 August 2022).

The RPE is composed of a monolayer of pigmented cells, which together with Bruch’s membrane, form the blood–retinal barrier [3]. With the exception of RPE microvilli, there are no intercellular junctions between the RPE and photoreceptor layers. In fact, the interphotoreceptor matrix (IPM) occupies the space between them and is responsible for the photoreceptor–RPE cell interactions and for the exchange of metabolites [4,5]. RPE cells are the most active phagocytic cells in the human body [6], and are also responsible for the nourishment of photoreceptors, amongst the nourishment of others too [3]. Therefore, a RPE loss results in a disruption of the photoreceptor’s feeding and consequently also photoreceptor death, leading to progressive vision loss and, ultimately, blindness.

The *RPE65* (HGNC: 10294; NM_000329.2) gene was one of the first RPE-expressed genes discovered to be associated with retinal disease [7]. The RPE65 protein is involved in the visual cycle, by converting all-*trans*-retinol produced by the photoreceptors into 11-*cis*-retinal, which is the active ligand for opsins in the outer segment of the photoreceptors [8]. The presence of pathogenic variants impairs the function of the RPE65 protein, causing an accumulation of retinyl esters in RPE cells. This, at the same time, leads to a reduction in the 11-*cis*-retinal levels, resulting in the photoreceptor cell death [8]. Currently, more than 138 pathogenic variants in the *RPE65* gene have been associated with different subtypes of IRDs such as retinitis pigmentosa, severe early childhood onset retinal dystrophy and Leber congenital amaurosis (LCA) [9,10]. LCA is one of the most frequent IRDs in childhood characterized by retinal dystrophy that appears commonly before the first year of life [11]. Its prevalence is in around 1:80,000 people worldwide [12]. It is estimated that between 4 and 16% of all LCA cases are caused by variants of *RPE65*, which frequency is more prevalent in Indian and Caucasian populations than in others like the Chinese population [13,14,15,16].

Around 15% of all IRD-causing variants affect the pre-mRNA splicing process, leading to exon elongation, exon skipping, intron retention or pseudoexon insertion [17,18]. Understanding the splicing defect is particularly relevant since the splicing modulation using antisense oligonucleotides is a potential therapeutic strategy for these types of variants and therefore a possible treatment for IRDs [1]. In the case of *RPE65,* there is already a gene augmentation approach using adeno-associated viruses on the market (Luxturna^®^ [19]).

Here, we investigated the pathomechanism of the c.11+5G>A variant in *RPE65.* Using several models, including patient-derived cells, we were able to conclude that the c.11+5G>A variant does not cause a splicing defect as was previously thought. Instead, it reduces the gene expression of *RPE65* in a cell-specific manner, highlighting the importance of using different cellular models to assess the effect of pathogenic variants.

## 2. Materials and Methods

### 2.1. Ethical Statement

This study followed the Declaration of Helsinki. Protocol 2018-4516 was approved from the local ethics committee of Arnhem-Nijmegen, the Netherlands. The patient material was obtained upon their signed informed consent.

### 2.2. Cell Lines

The human RPE cell line (ARPE19; ATTC #CRL-2302) and the human telomerase-immortalized RPE cell line (hTERT-RPE1; ATTC #CRL-4000) were cultured in Dulbecco’s Modified Eagle medium (DMEM)/Ham’s F12 medium (1:1 *v*/*v*) (Sigma-Aldrich, Saint Louis, MO, USA). Before mixing both mediums, Ham’s F12 was supplemented with 1% of Alanine-Glutamine (Sigma-Aldrich). Human embryonic kidney 293T (HEK293T; ATTC #CRL-3216) cells were cultured in DMEM. Both DMEM/F12 and DMEMs were supplemented with 10% fetal calf serum (Sigma-Aldrich), 1% sodium pyruvate (Sigma-Aldrich) and 1% penicillin-streptomycin (Sigma-Aldrich). The human retinoblastoma cell line (Weri-Rb1; ATTC #HTB-169) was cultured in Roswell Park Memorial Institute (RPMI) 1640 Medium (Sigma-Aldrich) supplemented with 15% FCS, 2% HEPES (Sigma-Aldrich) and 1% penicillin-streptomycin (Sigma-Aldrich). The cells were grown at 37 °C and 5.0% CO_2_.

### 2.3. In Silico Studies

The effect of the c.11+5G>A variant in the Leiden Open Variation Database (LOVD) was predicted as pathogenic or likely pathogenic. Furthermore, the effect on splicing, the presence of a non-canonical splice site and near exon aberrant RNA (NEAR) variants were predicted using the algorithms of SpliceSiteFinder-like (SSFL), MaxEntScan (MES), NNSPLICE (NSS) and GeneSplicer (SC) software, all of them embedded in the Alamut Visual software version 2.13 (Interactive Biosoftware, Rouen, France; http://www.interactive-biosoftware.com (accessed on 28 September 2022)). The SpliceAI prediction was also conducted indicating “raw” as a score type, and “500 nt” as a maximum distance (Illumina, San Diego, CA, USA; https://spliceailookup.broadinstitute.org accessed on 28 September 2022).

### 2.4. Generation of Vectors

All constructs employed in this work were generated using the Gateway cloning system (Thermo Fisher Scientific, Waltham, MA, USA) [20,21]. To generate the splicing vectors, the patients’ DNA was used as a template to amplify the *RPE65* gene from 5′ UTR to intron 3 using attB-tail primers (Appendix A) and cloned inside the entry clone attP-pDONR™201 plasmid (Invitrogen, MA, USA) following the manufacturer’s instructions. The PCR reaction was conducted using the Accuprime High Fidelity reaction kit (Thermo Fisher Scientific) supplemented with 0.3% of DMSO. The PCR conditions were set as follows: an initial denaturation at 98 °C for 30 s, followed by 35 cycles of melting (98 °C for 30 s), annealing (58 °C for 30 s) and extension (72 °C (1 kb/min)), with a final extension at 72 °C for 10 min. To generate the wild-type entry clone, a site-directed mutagenesis was conducted using the Phusion high-fidelity polymerase kit (Thermo Scientific, MA, USA) and the mutagenesis primers indicated in Appendix A. The site-directed mutagenesis was conducted according to the manufacturer’s instructions. The parental plasmid was digested with *Dpn*I (New England Biolabs, Ipswich, MA, USA). The resulting product was transformed in DH5α competent bacteria. All clones were verified by Sanger sequencing. Finally, the *RPE65* region of interest was introduced in the destination vector pcDNA3 by an LR-reaction.

To generate the luciferase-reporter assay vectors, both wild-type and mutant entry vectors were employed to clone the *RPE65* sequence into the pGL3-enhancer destination vector, resulting in the XL luciferase constructs by an LR reaction. All the shorter constructs (XS, S, M and L) were generated by a site-directed mutagenesis PCR, employing primers that deleted part of the *RPE65* sequence without affecting the rest of the entry vector sequence (Appendix A). To do that, a forward primer presented a tail which is complementary to the 10 first 5′-end nucleotides of the reverse primer and vice versa. For the XS, construct primers were designed to keep the transcription start codon in the final plasmid, but the c.11+5 position was removed. This construct was employed as the control. The PCR conditions were settled as indicated above for the site-directed mutagenesis. The absence of undesired mutations was confirmed by Sanger sequencing.

### 2.5. Midigene Splicing Assays

A total of 400,000 cells of HEK293T, hTERT-RPE1 and ARPE19 were transfected with the expression vectors or with the entry clones using FuGENE^®^-HD (Promega, Madison, WI, USA). For all the transfections, a 3:1 FuGENE^®^-HD Transfection Reagent:DNA ratio was maintained and the FuGENE^®^-HD/DNA mixture was delivered via an Opti-MEM reduced serum medium (Gibco, Waltham, MA, USA). Two million WERI-Rb1 cells were transfected following the same FuGENE-HD^®^:DNA ratio, but the transfection mixture was incubated for 2 h at 37 °C before plating them. In all cases, cells were harvested 48 h after transfection. Then, cells were subjected to RNA isolation using the NucleoSpin RNA kit (Machery-Nagel, Allentown, PA, USA). One microgram of the total RNA was retrotranscribed into cDNA using the kit iSCRIPT (Bio-Rad, Hercules, CA, USA). Both processes were conducted following the manufacturer’s protocol. Subsequently, the transcripts were amplified by a PCR with primers located in exons 1 and 3 of the respective constructs. As a loading control, *ACTB* was amplified. All mixtures of the PCR reaction (total 25 μL volume) contained 50 ng of cDNA plus 0.2 μM of each primer pair, 1 U Taq DNA Polymerase (Roche, Switzerland), 1X PCR buffer with MgCl_2_, 1× Q-Solution, 1 mM MgCl_2_ and 0.2 mM dNTPs (QIAGEN, Hilden, Germany). The PCR program included a denaturation step of 94 °C for 3 min, followed by 35 cycles of melting (94 °C for 30 s), annealing (58 °C for 30 s) and extension (72 °C for 1 min) steps, with a final elongation step of 72 °C for 10 min. The primer information is included on Appendix A. Finally, the semi-quantification of the different transcripts was performed using Fiji software (developed by National Institute of Health, Bethesda MD, USA and LOCI at University of Wisconsin, Madison, WI, USA) [22].

### 2.6. Luciferase Assays

The different pGL3-enhancer constructs were transfected into HEK293T and hTERT-RPE1 using FuGENE-HD^®^ Transfection Reagent as previously indicated. The pRL-CMV Renilla luciferase reported plasmid (Promega) was co-transfected for the internal control of cell transfection using the Dual-Luciferase reporter assay system kit (Promega), according to the manufacturer’s instructions. Firefly and Renilla luminescence were measured 48 h after the transfection using a luminometer (Berthold Sirius Single Tube Luminometer, Berthold Technologies (Bad Wildbad, Germany)). The Luciferase measurement was performed in duplicate using 5 µL of the cell lysate. The fold-change induction was estimated as the ratio of the Firefly between the Renilla average values. The data were normalized against the XS construct.

### 2.7. Induced Pluripotent Stem Cell (iPSC) Differentiation into Photoreceptor Precursor Cells (PPCs)

The SCTCi016-A iPSC line, previously generated and characterized in our lab [23], carries the c.11+5G>A variant in *RPE65* in homozygosity. As a control, we used the iPSC line generated and characterized at the Stem Cell Technology Center of the Radboudumc (iPSC15-00001). Both iPSC lines were kept in a culture growth-factor-reduced Matrigel (Corning) in an Essential 8 Flex medium (Thermo Fisher Scientific). The cells were passaged as clumps (ratio of 1:5–1:10) every 5–6 days. To obtain photoreceptor precursor cells (PPCs), the iPSC lines were seeded as single cells in 12-well plates, following the differentiation protocol previously described [24,25,26,27]. On day 29, one of the wells was treated with 100 µg/mL of cycloheximide solution (CHX, Sigma-Aldrich) (+CHX), while the other well was kept non-treated (-CHX). On day 30, the cells were rinsed with PBS, scraped and harvested for the RNA analysis. 

### 2.8. Induced Pluripotent Stem Cell (iPSC) Differentiation into RPE Cells

Differentiation from the patient/control iPSCs to the RPE cells (iPSC-RPE) was performed following the protocol of Regent et al., 2019 [28]. The differentiated cells became more pigmented and were selected and passed to obtain pure RPE cultures. During all the differentiation processes, the medium was changed every 2–3 days. All experiments were performed in P3 RPE cells and the experiments were performed in duplicate (independent differentiations). One day before harvesting, the RPE cells were treated with either 100 µg/mL or 200 µg/mL of CHX (+CHX 1× or +CHX 2×, respectively), while the other well was kept non-treated (-CHX). 

### 2.9. Quantitative PCR (qPCR)

The RPE markers were analyzed by qPCR. In this case, the RNA isolation of the iPSCs (day 0) and RPE cells was performed as described above. Then, one microgram of total RNA was retrotranscribed employing a SuperScript VILO Master Mix (Thermo Fisher Scientific) according to the manufacturer’s protocol. A qPCR reaction was conducted using the GoTaq Real-Time qPCR Master kit (Promega) and the samples were processed in an Applied Biosystem QuantStudio 5 Digital system. The expression levels of the RPE markers were normalized against the housekeeping gene (*GUSB*). The employed primers are listed in Appendix A. Each sample was normalized against the expression of the housekeeping gene and compared to the iPSC (day 0) using the 2^−(ΔΔCt)^ method [29]. 

### 2.10. Immunochemistry Assays

The P3 iPSC-RPE cells were grown on coverslips. The cells were rinsed with 1× PBS, fixed in 4% paraformaldehyde for 10 min at 4 °C and then permeabilized in PBS supplemented with 1% of Triton X for 5 min at RT. The cells were cleaned in 1× PBS and subsequently blocked in PBS supplemented with 2% of bovine serum albumin for 30 min at RT. For the immunostaining, the cells were incubated with the primary antibody diluted in a 2% bovine serum albumin in 1× PBS for 2 h at RT. The cells were washed 4 times for 5 min in 1× PBS and incubated between 45 and 60 min with the corresponding secondary antibodies. The cells were washed 3 times for 5 min in 1× PBS. Finally, the slides were mounted in Vectashield without DAPI (Life Technologies, Carlsbad, CA, USA). The cells were imaged on a Zeiss Axio Imager Z1 Fluorescent microscope (Zeiss, Aalen, Germany) and analyzed using Fiji software. The employed antibodies and their combinations were indicated in Appendix A. 

### 2.11. Cell-Size Measurement

Immunochemistry images were analyzed by Wimasis Image analysis (Onimagin Technologies SCA, Córdoba, Spain). This on-line facility estimated the number of nuclei and the cell area of three independent staining for the control and patient iPSC-RPE cells. The data were subsequently employed for the calculating of the mean of each staining first, and then the average of the control and patient iPSC-RPE.

### 2.12. Western Blot

The P3 iPSC-RPE cells were harvested and homogenized in 150 µL of RIPA buffer (50 mM Tris-HCl pH 7.5, 150 mM NaCl, 1% NP40, 0.1% SDS, 0.5% Sodium deoxycholate and 1 mM EDTA) supplemented with proteinase inhibitors (cOmplete^TM^ ULTRA tablets; Roche, Basel, Switzerland). The protein concentrations were quantified using the BCA Protein Assay Kit (Thermo Scientific) according to manufacturer’s instructions. A total of 35 µg of protein from the control and patient-derived RPE cells was diluted in NuPage loading buffer with 10% of DTT (Invitrogen, Waltham, MA, USA) and run in 4–15% MINI-protean TGX Stain-free gel (#4568084, Bio-Rad) at 200 V for 37 min. The proteins were transferred to a nitrocellulose membrane (trans-blot Turbo transfer pack, Bio-Rad) by using the Trans-blot turbo transfer system (Bio-Rad), according to the manufacturer’s recommendations. The membranes were blocked with a WestVisionTM diluent (Vector Laboratories, Newark, CA, USA) for 1 h at room temperature (RT) and subsequently incubated with the primary antibodies (Appendix A) overnight at 4 °C. After that, the membranes were washed 3 times with PBS 0.1% Tween-20 for 5 min, incubated with a secondary antibody (Appendix A) for 1 h at RT and then washed 3 times in PBS with 0.1% Tween-20 for 5 min. The blots were developed in the Odyssey Imaging System (Li-Cor Biosciences, Lincoln, NE, USA). The detected bands were semi-quantified using Fiji Software [22]. The band’s quantification was conducted twice in each replicate and the values were normalized against Histone H3. 

### 2.13. Immunoprecipitation

The control iPSC-RPE cells were lysed in 200 μL of ice-cold RIPA lysis buffer supplemented with proteinase inhibitors by cell scraping. Once the protein was isolated, its concentration was measured by BCA as indicated previously. Thirty-five micrograms of the total lysate were reserved for the Western blot analysis, performed as described in a previous section. The remained lysate was incubated with the primary antibody against RPE65 (Appendix A) overnight on a rotating wheel at 4 °C. A total of 40 μL of Protein A/G PLUS-Agarose beads (Santa Cruz technologies, TX, USA) were washed in a cold RIPA buffer to conduct a pre-clearing of the lysate (1 h on a rotating wheel at 4 °C) and the subsequent incubation (2 h on a rotating wheel at 4 °C). After that, the input samples were collected and the beads were cleaned 3 times for 5 min in a cold RIPA buffer to be finally resuspended in 50 μL of loading buffer (1× loading dye, 100 mM DTT, diluted RIPA buffer). From them, 45 μL of the sample were loaded in NuPAGE 4–12% BisTris gel (Life Technologies). Then, the gel was rinsed with deionized water 3 times for 5 min at room temperature to be subsequently stained with SimplyBlue SafeStaining (Invitrogen) for 1 h at RT with gentle shaking. The bands of interest were cut and stored at −20 °C for a subsequent proteomics analysis. The remained 5 μL of the sample obtained from the immunoprecipitation protocol were analyzed by a standard Western blot.

### 2.14. In-Gel Digestion and Proteomic Analysis

The digestion for the cut bands to isolate the protein for the mass spectrometry analysis was based on a previous publication [30]. Briefly, the bands were excised in small fractions to remove the excess of the gel. Then, the gel pieces were washed, and neat acetonitrile was added to shrink the pieces. After discarding the supernatant, the samples were reduced using 10 mM of DTT for 30 min and subsequently alkylated with 50 mM of 2-chloroacetamide for 30 min in the dark. The samples were washed again and digested overnight using trypsin (Promega). The samples were diluted in miliQ with 0.1% FA before the injection. 

The samples were injected into a nano-HPLC (nanoElute, Bruker Daltonics, Billerica, MA, USA) with a one-column separation coupled to a timsTOF Pro 2 (Bruker Daltonics). The instrument was equipped with the CaptiveSpray source (Bruker Daltonics). The samples were separated by liquid chromatography using a C18 analytical column (nanoElute FIFTEEN, Bruker Daltonics, 150 mm length, 75 µm I.D, 1.9 µm particle size) at 45 °C with a flow rate of 500 nL/min. The liquid phase consisted of water (Buffer A) or acetonitrile (Buffer B) supplemented with 0.1% formic acid (Biosolve, Valkenswaard, the Netherlands) and 0.01% trifluoroacetic acid (Sigma Aldrich). A linear gradient was used from 5 to 43% buffer B in 25 min. TimsTOF pro 2 was operated in positive (Parallel Accumulation-Serial Fragmentation) PASEF mode. MS and MS/MS spectra were acquired with a mass range of 300–1800 *m/z*, with a mobility range of 0.6–1.8 K0 and with a ramp and accumulation time of 100 ms. The data-dependent acquisition was performed using 10 PASEF MS/MS scans per cycle with a 100% duty cycle, a prepulse storage time of 12.0 µs and a transfer time of 60 µs. An active exclusion time of 0.4 min was applied as well as precursors that reached 20,000 intensity units. The collision cell RF was set to 1500 Vpp and the collision energy was ramped as a function of the ion mobility. 

### 2.15. Analysis of Proteomic Data

The LC-MS/MS datasets were converted to mascot generic files (MGF) using an in-house script in DataAnalysis (Bruker Daltonics v5.3). The analysis of the MGF files was done by using MSFragger (v3.4, Nesvilab, Ann Arbor, MI, USA; https://msfragger.nesvilab.org/ accessed on 28 September 2022) coupled to the interface Fragpipe (v17.1, Nesvilab). For all the searches, a database of the protein sequence of reviewed human protein was used (Uniprot, 20,361 entries). A decoy database was generated and added to this database. The precursor mass tolerance and fragment mass tolerance were set to 20 ppm. The tryptic cleavage specificity was set with a maximum of 2 missed cleavages. The variable modifications were set for methionine oxidation, protein N-terminal acetylation, phosphorylation and pyroglutamate. The fixed modifications were set for carbamidomethyl cysteine modifications. The allowed peptide length is 4–50 residues and 500–5000 Da. PeptideProphet in Philosopher (v4.1.1, Nesvilab) was set in closed search settings for peptide-spectrum matches (PSM) filtering. A label free quantification was run with IonQuant (v1.7.17, Nesvilab) with a false discovery rate (FDR) of 1% for both the peptides and proteins. The ion mobility tolerance was set for 0.05 1/k0 and normalize was enabled. The processed data were exported to Microsoft Excel for a visualization.

### 2.16. PACBIO Long-Read Genome Sequencing and Analysis

Amplicon libraries were prepared based on PacBio’s protocol “*Procedure & Checklist*—*Preparing SMRTbell libraries using PacBio Barcoded Adapter for multiplex SMRT Sequencing*” (PACBIO, Pacific Biosciences, Menlo Park, CA, USA). Then, the binding between the sequence primer and polymerase was based on the recommendations in SMRTlink v8.0 (PACBIO; https://www.pacb.com/support/software-downloads/ accessed on 28 September 2022).

The data analysis was conducted as follows: small variant (SNV and indel) calling was performed using Google DeepVariant v1.1 (https://github.com/google/deepvariant accessed on 28 September 2022) [31] and WhatsHap v1.1 (https://github.com/whatshap/whatshap accessed on 28 September 2022) [32]. The variant comparison between the samples was performed using VCFTools v0.1.13 [33] across both the genome and exome (including splice sites). The analysis of the structural variants was conducted by utilizing a joint calling on PBSV v2.4.0 (Pacific Biosciences, Menlo Park, CA, USA; https://github.com/PacificBiosciences/pbsv accessed on 28 September 2022) and retaining the ‘PASS’ calls.

### 2.17. Statistical Studies

GraphPad Prism 9.0 software (GraphPad, San Diego, CA, USA) was employed for the statistical analysis. The Luciferase-reporter assays are represented as the mean ± standard deviation (SD). The data were normalized against the XS construct. The differences between the wild-type and mutant constructs were assessed by a one-way ANOVA test followed by a Bonferroni correction. *p*-values equal to or smaller than 0.005 were considered to be statistically significant. Differences in the size between the control iPSC-RPE and patient iPSC-RPE were assessed by a two sample *t*-test after Welch’s correction; *p*-values equal to or smaller than 0.005 were considered to be statistically significant. 

## 3. Results

### 3.1. The c.11+5G>A Variant of RPE65 Results in Exon Elongation In Vitro

We selected the c.11+5G>A due to its high recurrence and prevalence (87 alleles reported and third most prevalent variant according to the LOVD database, last access 24 August 2022). In LOVD, this variant is classified as pathogenic or likely pathogenic, especially with a recessive inheritance. This designation was based on the genetic classification of the American college of medical genetics (AMCG) criteria. In our in silico analysis, two of the algorithms (SSFL and GS) in Alamut visual predicted a loss of the canonical splice donor site of exon 1, while the other two algorithms (MES and NNA) showed a dramatic reduction in the score (Figure 1). A cryptic splice donor site located 124 nt downstream was foreseen by one of the algorithms with similar scores as the mutated original splice donor site. This prediction was validated by using SpliceAI, which *delta score* also showed a reduction in the canonical splice donor site and an increment of the value of the cryptic donor site located at 124 nt downstream from it (Figure 1). Altogether, these in silico studies pointed towards a potential splicing defect in exon 1, namely with a 124 nt exon elongation.

To demonstrate whether this variant altered the splicing, we designed a midigene splicing assay. Midigenes have been very useful to predict splicing defects in other genes [34,35]. However, these midigenes are flanked by reporter exons that contribute to facilitate the detection. Since the variant of the study was in exon 1, we cloned the genomic region of *RPE65* from the 5′ UTR to intron 3 in a pcDNA3 vector under the control of a CMV promoter. Thus, the insert will be transcribed and the splicing can be assessed by RT-PCR upon a transfection in conventional cells. The splicing was assessed in four different human cell lines: HEK293T, Weri-Rb1, ARPE-19 and hTERT-RPE1. In all cell lines, with the exception of Weri-Rb1, the presence of the variant resulted in an exon elongation (Figure 2A) as predicted in silico. Sequencing analysis revealed a 124 nt exon elongation, resulting in an out-of-frame transcript, with a premature stop codon and therefore potentially degraded by a nonsense-mediated decay (NMD).

### 3.2. The c.11+5G>A Variant-Mediated Splicing Defect Is Less Noticeable in Patient-Derived iPSCs and Photoreceptor Precursor Cells

To gain full insights into the molecular context, patient-derived cells were reprogrammed to induced pluripotent stem cells (iPSCs). This line, which harbors the variant c.11+5G>A in homozygosis, was fully characterized previously [23], and was used to generate photoreceptor precursor cells (PPCs). These patient-derived models allowed the study of the possible NMD effect by inhibiting this process by adding cycloheximide (CHX) to the medium. In both, the 124 nt exon elongation transcript was also detected in the patient-derived cells (Figure 2B). As expected, this out-of-frame transcript is degraded by NMD as the detection increases upon a CHX exposure. Besides that, in iPSCs, the correct amplicon was also not detected in the absence of CHX, which could be explained by a limitation in the detection due to the very low expression of *RPE65* in this cell type. Remarkably, however, the levels of the exon elongation in these models were too low to explain a pathogenic mechanism in a recessive disorder, as the wild-type product was clearly more than 50%.

### 3.3. The Variant c.11+5G>A reduces mRNA Expression Resulting in RPE65 Absence in RPE Cultures

To study this observation in more detail, we used the patient-derived iPSC line to generate the RPE cells, in which *RPE65* is highly expressed. In contrast to previous results, the aberrant transcript with the 124 nt exon elongation was barely detected (Figure 2C). Intriguingly, however, the expression levels of the wild-type *RPE65* transcript were much lower compared to the control RPE cell line, even after inhibiting NMD. These results were consistent in several regions of *RPE65* cDNA that were amplified (Figure 3). Contrarily, no differences were observed when *BEST1*, another RPE-specific transcript, was amplified (Figure 3).

To further characterize these RPE cells, we assessed the expression of RPE markers (RPE65, BEST1 and EZRIN) at the protein level by immunohistochemistry (ICC) in both the control and patient-derived RPE (Figure 4). No differences between BEST1 and EZRIN staining were detected between the control and patient-derived RPE cells. However, the patient-derived RPE did not show staining for the RPE65 protein, while a clear signal was observed in the control-derived RPE cells (Figure 4), which was in line with the RNA analysis. Next, we assessed the protein levels by Western blot (Figure 5). The RPE65 protein was only detected in the control-derived RPE, but not in the patient-derived RPE cells (Figure 5A,B), confirming the absence of detectable levels of the RPE65 protein in the patient-derived RPE cells. No differences were observed for MERTK, which was used as a RPE marker (Figure 5A,C). These results are also in line with the RNA analyses in which the RPE markers did not show differences, and the RPE65 levels were dramatically decreased (Figure 6A).

Surprisingly, a prominent protein of about 35 kDa was also detected by Western blot (Figure 5A). This band was also detected in the RPE65 positive control (bovine RPE), although its intensity in both the control and patient-derived RPE was significantly higher. To elucidate the nature of this band, an immunoprecipitation assay using the RPE65 antibody was conducted in fresh control RPE cells, followed by a proteomic analysis. After applying the filtering steps and removing the epidermal/keratin proteins or broadly expressed proteins detected for both bands, the analysis showed 13 out of 91 proteins were left as suitable candidates for the ~35 kDa unknown protein. From these, only two proteins, GTR1 and ALDOC, have a specific expression in the retina, but only ALDOC was detected in the input of the ~35 unknown kDa protein band (Appendix A), albeit at low levels. Another possibility could be that this ~35 kDa protein corresponds with a previously non-identified isoform of RPE65, or a processed RPE65 protein. When analyzing the peptides sequences, this analysis suggests that this is not the case due to the broadly, but not continuous, distribution of the peptide coverage along the RPE65 peptide sequence (Appendix A).

As mentioned before, the gene expression analysis of the RPE-specific markers by RT-PCR (BEST1, Figure 3) or qPCR (BEST1, MERTK and DCT, Figure 6A), as well as the detection of the RPE specific proteins by ICC (EZRIN, BEST1, Figure 4) or Western blot (MERTK, Figure 5), did not show differences between the control and patient cells, indicating comparable differentiation stages. Despite these molecular results, we observed some consistent discrepancies between both RPE lines. First, the intensity of the cell pigmentation of the patient-derived RPE was less than the control line at the same time point (Figure 6B). Even when the cells were cultured for longer times, these cells never became as pigmented as the control lines. In addition, the patient-derived RPE cells seemed to be bigger in size than the control-derived RPE cells. This was visible in bright-field microscopy and in ICC by staining with ZO-1, a tight junction marker (Figure 4 and Figure 6B,C). To confirm this observation in a quantitative manner, we measured the area of the cells delimited by the ZO-1 staining (tight junctions). The results indicated that the patient-derived RPE cells were 1.5 times bigger than the control-derived RPE cells.

### 3.4. Long-Read Genome Sequencing Does Not Point Other Causative Variant in The Patient-Derived DNA

Given the low expression of *RPE65* at the RNA level and the differences observed also in the iPSCs and PPCs, we performed long-read PacBio sequencing to discard any possible rearrangement that could explain the different results observed in the different lines. We sequenced the patients’ DNA obtained from the blood and DNA obtained from the iPSCs cells. The obtained results discarded any possible rearrangement due to the reprogramming of the cells [23]. We also assessed the region of interest and we could not detect any deletion, insertion or other type of rearrangement. We noticed that no SNPs were present in heterozygosis in the entire region, however, the coverage of *RPE65* was comparable to other genes and other sequenced samples, discarding a potential deletion (data not shown). Overall, we could not identify any other possible genetic cause that could explain the reduction in the expression of *RPE65*.

### 3.5. The c.11+5G>A Variant-Mediated mRNA Expression Reduction Is Cell-Context Dependent

Lastly, to confirm the effect of the c.11+5G>A variant in the gene expression using an alternative method, we generated different luciferase reporter constructs. These vectors had different lengths and included the 339 nt upstream region of *RPE65* prior to the 5′UTR region (Figure 7A). To assess the activation of the expression, these constructs were transfected into HEK293T and hTERT-RPE1 cells. The studies in HEK293T cells revealed no differences between the wild-type and mutant reporter vectors for all the constructs tested (Figure 7B). However, in the hTERT-RPE1 cells, all the mutant constructs presented a consistent and significant reduction in the luminescence in comparison with the wild-type condition (Figure 7C), indicating that the variant has a detrimental effect on the regulation of the expression of *RPE65* in an RPE-specific context. The fact that this effect was observed in all constructs supports the hypothesis that this downregulation is triggered by the c.11+5G>A variant.

To further validate these results, entry clones (which only include the endogenous *RPE65* promoter) were transfected into different cells lines (HEK293T, ARPE19, hTERT-RPE1 and Weri-Rb1) to study the differences in the expression levels. In line with previous observations, only the HEK293T cells present similar levels in the wild-type transcript between wild-type and mutant entry clones, while the rest of the cell lines (which had a retinal origin) showed reduced levels of the expression of the wild-type transcript after transfecting the mutant entry vector. In addition to that, only the HEK293T cells transfected with the mutant entry vector showed the out-of-frame 124 nt transcript, which is indiscernible from the other cell lines transfected with the same plasmid (Appendix A).

## 4. Discussion

In recent years, the importance of variants affecting pre-mRNA splicing as the cause of retinal disorders has been recognized [36,37,38,39]. These usually lead to aberrant transcripts with detrimental effects on the protein levels and, therefore, the cell function [40]. However, evidence is also emerging that splicing defects can occur in a tissue- or even cell-type manner [41,42]. In addition, bioinformatic splicing predictions are not always accurate, highlighting the importance of studying potential splicing-affecting variants in the relevant model systems.

In the present work, we investigated an intronic variant classified as pathogenic or likely pathogenic that, according to the in silico predictions, would have an effect on the splicing. Unexpectedly, our results indicated that this effect is highly dependent on the molecular and cellular context. Employing the midigene systems, iPSC or PPCs, we observed an exon elongation of 124 nt as predicted in silico in part of the transcripts, while in the RPE cells, in which *RPE65* is endogenously expressed, this variant resulted in a significant reduction in the gene’s expression. In all cases, the wild-type transcript was still detectable by RT-PCR. These results suggest that both the splicing defect and the alteration in expression do not have a fully penetrant effect, at least under the conditions tested in this work. This has been already observed for several other variants, in which the splicing defect was present in the majority but not in all of the transcripts [25,27].

A role of intron 1 of *RPE65* in the regulation of the gene has been suggested, but only poorly investigated [43,44]. According to Boulanger and colleagues, the c.11+5G>A variant is placed in a regulatory region that, in mice, accounts for a tissue-specific expression in the RPE [43]. In humans, the current knowledge about this region is even more limited. Recently, the presence of *cis*-regulatory elements in the upstream region (from 5′UTR to intron 4) of human *RPE65* were described. Interestingly, these elements are found only in the RPE/choroid tissue rather than the entire retina [45], supporting cell-specific consequences when they are affected. Despite that, to our knowledge, there is not any other study about how these elements regulate the expression of *RPE65* in humans. Therefore, either the disruption of an enhancer or the activation of a silencer by the c.11+5G>A variant could explain the expression defect observed in our study, and further research is still needed to elucidate the exact mechanism.

Previously, it has been reported that some *RPE65* missense variants (c.200T>G and c.430T>C), with no predicted effect in splicing, lead to a reduction in the expression of the mRNA of *RPE65* and barely detectable levels of the RPE65 protein in ICC [46]. However, no differences in cell size nor in the pigmentation levels were reported. We did not detect any RPE65 protein at all, and also observed clear differences in the pigmentation and cell size, despite the fact that selected differentiation markers showed equal expression levels at the RNA and/or protein level. These differences might be related to the severity of the variant and the total lack of the RPE65 protein. According to the literature, the pigmentation of iPSC-RPE can correlate to the maturation of the RPE [47]. Taking this assumption into account, we hypothesize that our patient iPSC-RPE cells could be less mature, despite being equally differentiated, than the control iPSC-RPE cells. In fact, the maturity of the RPE has been reported to affect not only the pigmentation [48] but also the morphology [49]. However, the maturation molecular signature of the iPSC-derived RPE has not been clearly established and there is still no consensus about the culture duration to obtain maturity in vitro [47]. In addition, the qPCR analysis of the RPE markers did not show statistical differences between our control and the patient line (Figure 6A), and this was also confirmed by ICC (Figure 4) and Western blot (MERTK, Figure 5). Therefore, the role of RPE65 in the maturation of the RPE and its effects at functional levels remain unclear and need to be further investigated. Independently of this, the absence of RPE65 will likely have a tremendous impact on the visual cycle. In this pathway, RPE65 is responsible for the conversion of all-*trans*-retinyl esters into 11-*cis-*retinal and free fatty acids. The disruption of this step would result in the toxic accumulation of all-*trans*-retinyl esters, as was previously reported in animal models [50]. In vivo, Sheridan et al. showed that RPE65 not only enables the formation of 11-*cis*-retinal but also facilities the availability of esters to be converted back to all-*trans*-retinol [50]. It is unclear yet whether it would be possible to observe defects in this pathway in iPSC-derived RPE, as only part of the pathway is represented in this cellular model.

In this study, we demonstrate the absence of RPE65 in our patient-derived RPE line by ICC and Western blot. In the latter, a band of ~35 kDa was detected in a high intensity both in the control and patient-derived RPE. This band did not match with any of the protein products from the RPE65 cleavage, which results in two proteins of 45 kDa and 20 kDa [51]. In addition, the epitope of the employed antibody is unknown, but previous reports studying known RPE65 epitopes suggested that some visual cycle proteins could be also detected [52]. Thus, in order to identify the nature of this ~35 kDa band, we conducted a proteomic analysis, which determined that this band was not related to RPE65. From the 13 suitable candidates, only ALDOC has been described in a retinal context. The expression in the retina of this glycolytic enzyme shows a good correlation with oxidative stress, reflecting the developmental and nutritional status of the tissue [53]. ALDOC is therefore a good candidate to explain this band, however, as we lack the entire 3D tissue context and corresponding interactions, we cannot discard other proteins which are not considered to be top candidates. Indistinctly of these results, the ~35 kDa band was detected at equal levels in both the control and patient-derived RPE cells, indicating that the pathogenic variant or lack of the expression of RPE65 did not have a significant effect on it.

Identifying and understanding the pathomechanism of genetic variation is crucial to develop therapeutic approaches. While for *RPE65*, a therapeutic approach (Luxturna^®^) based on a viral delivery is available, this is not the case for almost all IRD subtypes. Alternative approaches, in which the gene size is not limiting, are being developed. In particular, for defects caused by (deep-)intronic variants leading to splicing defects, antisense oligonucleotides have shown promising results at preclinical and, in some cases, also at clinical levels: *CEP290* [54,55,56,57,58,59,60], *ABCA4* [25,26,61,62], *USH2A* [63], *OPA* [64] or *CHM* [65]. Thus, it is important to identify the contribution to diseases of (deep-)intronic variants to ensure that proper treatments are developed. The use of different models along this study has allowed for the understanding of the main effect of the variant and its possible consequences with regard to the development of disease. We and others have shown how (deep-)intronic variants can lead to multiple outcomes depending on the model used [25,27,66]. Therefore, when possible, the closest cellular model to the real situation in vivo should be employed to study the effect of (deep-)intronic variants.

In summary, this work provides evidence that the homozygous c.11+5G>A variant in *RPE65* leads to a significant decrease in the expression in an RPE-specific context. Directly or indirectly, this seems to cause a delay in maturation, however, the mechanism remains unclear. Furthermore, our study also highlights the importance of choosing the relevant model systems to investigate potential splicing defects. The choice of the model, when possible, should be based on the most similar cell type to the one of where the gene of interest is expressed, to have a reliable understanding to what may happen in vivo.

## Figures and Tables

**Figure 1 cells-11-03640-f001:**
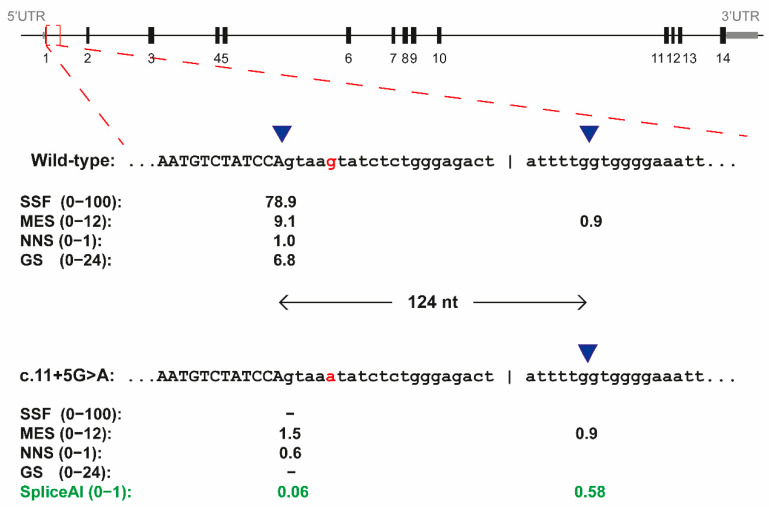
In silico splice-site predictions for the c.11+5G>A variant of *RPE65* by Alamut 2.13. Schematic representation of the *RPE65* gene and enlargement of the exon 1 and intron 1 for both the wild-type sequence and for the c.11+5G>A variant. Four in silico predictions of different algorithms are indicated below each sequence. The blue triangles indicated the position of splice donor sites, which are separate 124 nt. In red, the nucleotide on position c.11+5 is indicated. In green, the prediction by SpliceAI (*delta score*). SSFL, SpliceSiteFinder-like; MES, MaxEntScan; NNS, NNSPLICE and GS, GeneSplicer.

**Figure 2 cells-11-03640-f002:**
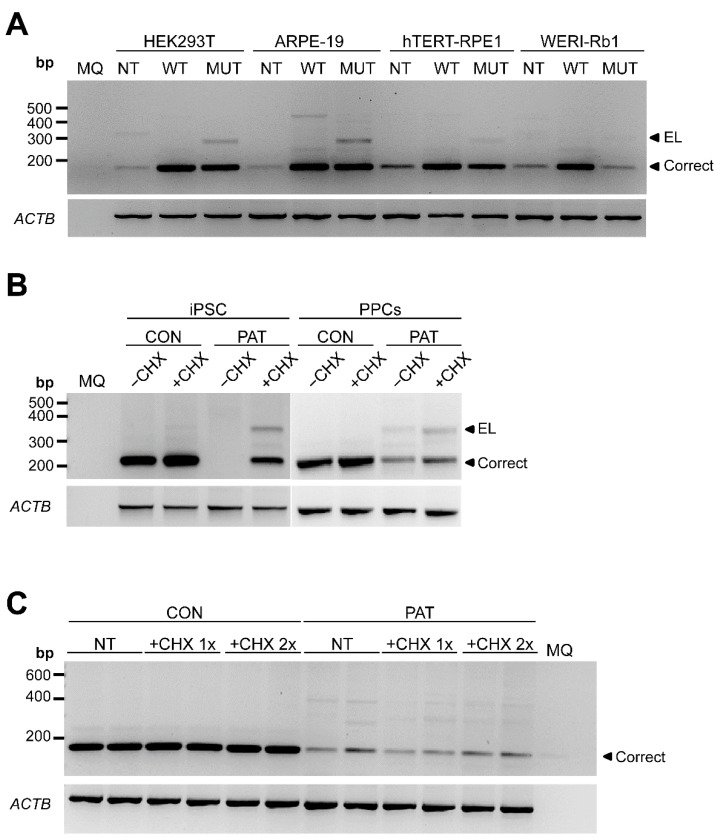
Analysis of *RPE65* gene expression by RT-PCR in different cellular models. (**A**) Midigene splicing assays in ARPE-19, WERI-Rb1, hTERT-RPE1 and HEK293T. Representative electrophoresis gel (n = 2) of the RT-PCR product after amplifying exon 1 to 3 of *RPE65* comparing wild-type (WT) and mutant (MUT) expression vectors. NT: non-transfected. (**B**) Representative gel of the RT-PCR of control (CON) and patient-derived (PAT) iPSC and photoreceptor precursor cells (PPCs). Cells were grown in absence (−CHX) or presence of cycloheximide (+CHX). (**C**) Representative gel of the RT-PCR of control (CON) and patient-derived (PAT) RPE cells. The cells are untreated (NT), treated with the normal amount of cycloheximide (+CHX 1×) or the double amount of cycloheximide (+CHX 2×). *ACTB* amplification was used as loading control. MQ: milliQ water; EL: exon-elongation; correct: wild-type transcript; bp: base-pairs.

**Figure 3 cells-11-03640-f003:**
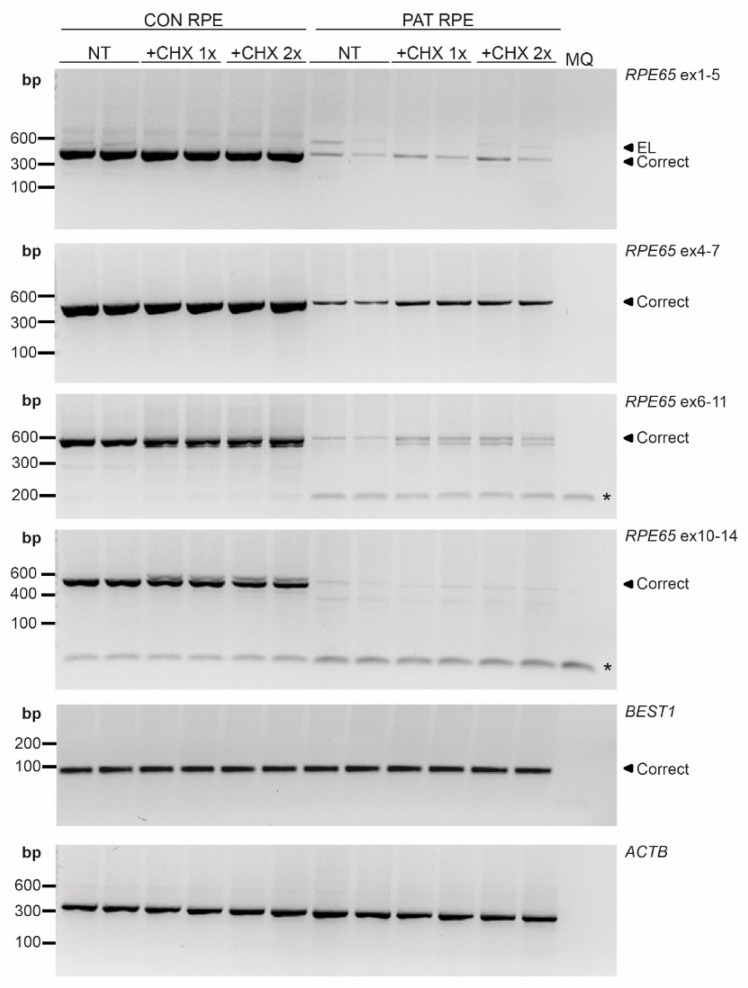
RT-PCR analysis in iPSC-derived RPE cells. Different regions of *RPE65* were amplified in non-treated RPE cells (NT) and RPE cells upon treating the cells with the normal dose (+CHX 1×) or double dose (+CHX 2×) of cycloheximide for both the control (CON) and patient (PAT RPE cells. All amplicons showed *RPE65* reduction in PAT RPE. * indicates primer dimers. *BEST1* was used as RPE marker detection and *ACTB* was used as a loading control. MQ: milliQ water; EL: exon-elongation; correct: wild-type transcript; bp: base-pairs.

**Figure 4 cells-11-03640-f004:**
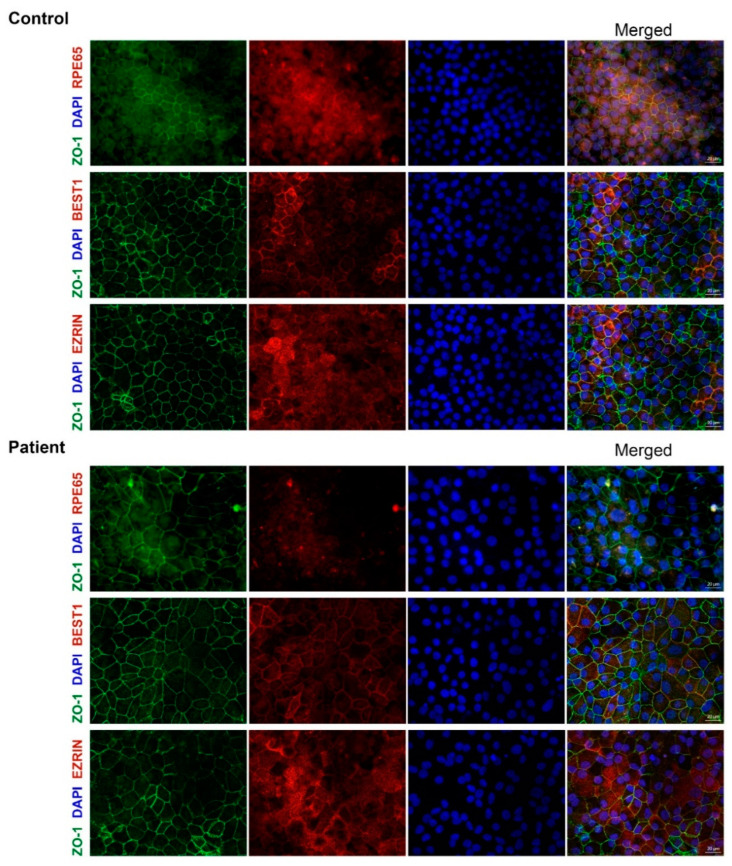
Immunocytochemical analysis of control and patient-derived RPE cells. P3 RPE cells were analyzed by ICC using several markers. In all case ZO-1 (green) stains the tight-junctions and DAPI (blue) stains the nuclei. In red, different RPE markers were assessed: RPE65 (first row), BEST1 (second row) and EZRIN (third row). Each channel is depicted individually and the merged is provided in the last column. Scale bar represents 20 μm.

**Figure 5 cells-11-03640-f005:**
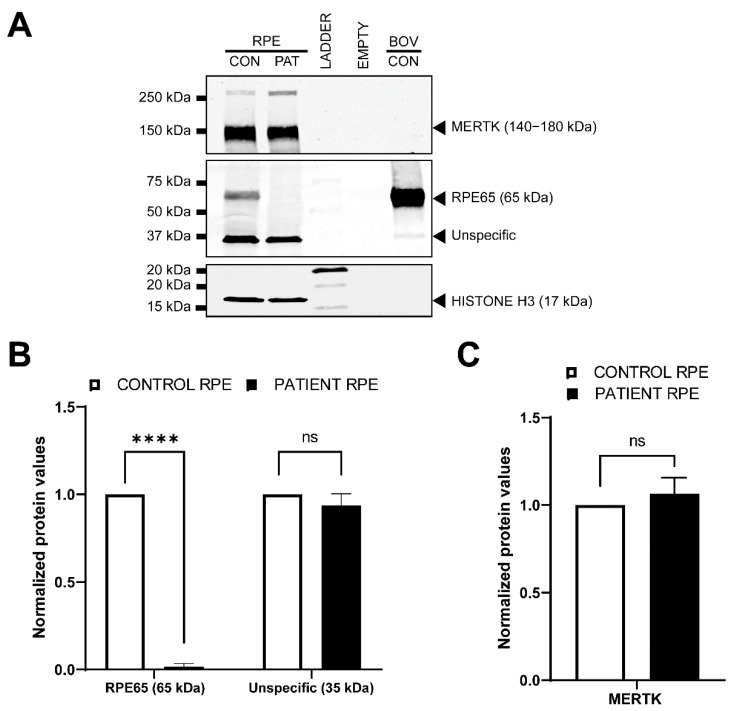
Western blot analysis in RPE cells. (**A**) Representative Western blot on P3 RPE cells. The upper panel depicts the immunodetection of the RPE marker MERTK. The middle panel shows the detection of RPE65, which expected size is 65 kDa. Bovine control (BOV-CON) was employed as a positive control. The lower panel represent HISTONE H3, which was used as loading control. Of note, MERTK and HISTONE H3 antibodies do not react with the bovine protein. (**B**) Quantification (n = 2) of the two bands detected on the middle panel (RPE65). Patient RPE cells present a significant reduction in RPE65 protein amount in comparison with the control RPE cells. There were no differences for the levels of the ~35 kDa protein levels. (**C**) Quantification (n = 2) of the protein levels of the RPE marker MERTK. Statistically differences by one-way ANOVA test are indicated as **** *p* < 0.00001 or ns: not-significant.

**Figure 6 cells-11-03640-f006:**
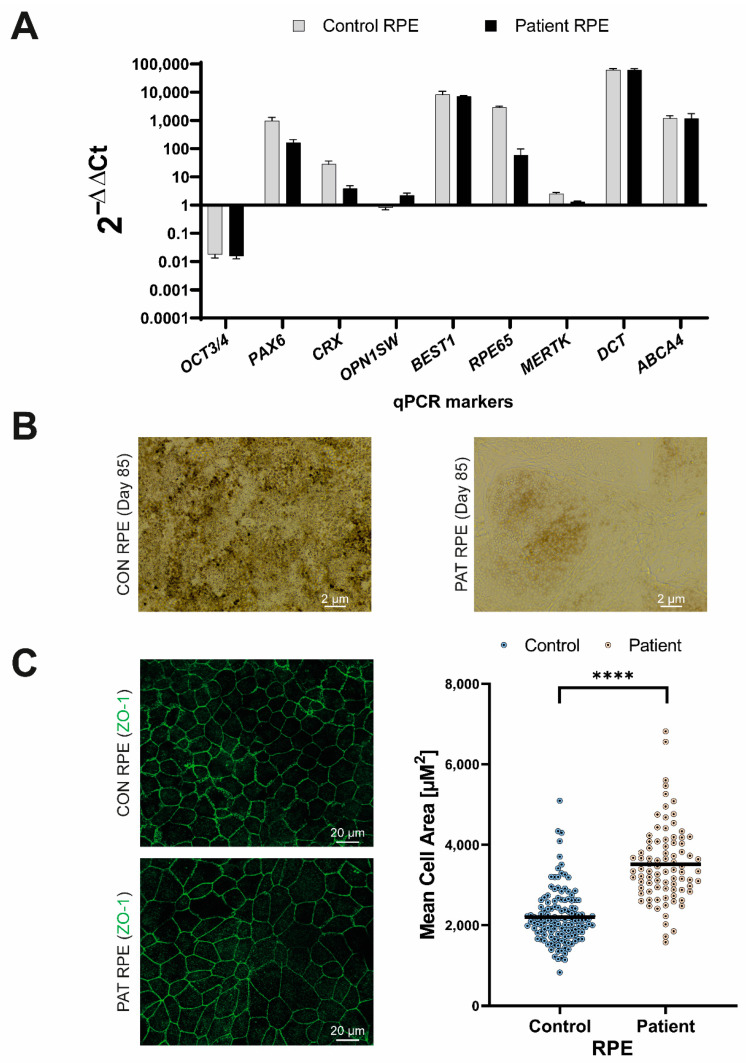
Characterization of the RPE differentiation. (**A**) Gene expression profile at the end of the differentiation process. P3 RPE cells were compared to iPSCs (day 0 of differentiation). The differentiation into RPE is observed by the expression levels of *BEST1*, *MERTK*, *RPE65* and *DCT*. (**B**) Bright-field image of the control (CON) and patient (PAT)-derived RPE at P3. Pigmentation is clearly observed in the CON RPE line. Scale bar represents 2 μm. (**C**) On the left panels, representative immunocytochemical images stained for ZO-1 (in green) which marks the tight junctions. Scale bar indicates 20 μm. On the right panel, scatter plot of the cell area (μM^2^) of every quantified cell based on the ZO-1 staining pictures using Wimasis Image analysis software. The average size for control and patient RPE cells is represented by a black line. Statistically differences by two sample T-test are indicated as **** *p* < 0.00001.

**Figure 7 cells-11-03640-f007:**
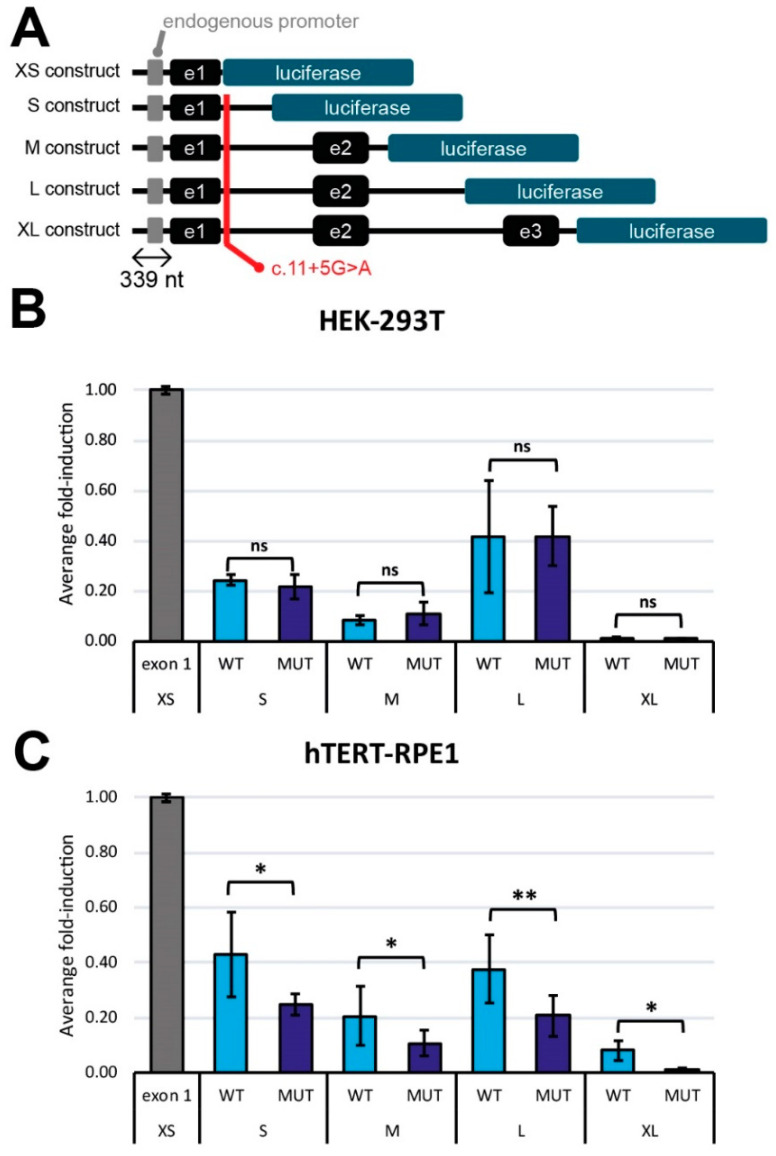
Expression analysis using luciferase assays. (**A**) Schematic representation of the different luciferase constructs and the position of the c.11+5G>A mutation. All constructs contain the 339 nt upstream region from the exon 1. (**B**,**C**) are a comparison of the average fold-induction between the wildtype transcripts (WT) and the ones harboring the mutation (MUT) in HEK-293T (**B**) and hTERT-RPE1 (**C**) (n = 4). Each bar represents the mean value in percentage ± SD. Statistical differences with respect the negative control (XS construct) by one-way ANOVA test are indicated as * *p* < 0.05 and ** *p* < 0.01 or ns: not-significant.

## Data Availability

Not applicable.

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
