# Peer review of "The Predicted Splicing Variant c.11+5G>A in RPE65 Leads to a Reduction in mRNA Expression in a Cell-Specific Manner"

_cells, 2022, doi:10.3390/cells11223640_

Round 1

Reviewer 1 Report

The authors focused the study on a relevant issue related  to inherited retinal diseases (IRDs), which is represented by a very  heterogenous group of neurodegenerative disorders. The outcomes of these diseases   inevitably lead to irreversible damage to photoreceptors and to retinal pigment epithelium cells. Since more than 250 genes are involved in the heterogenous variant of IRD, the aim of the authors was to investigate one of the variants in RPE65using several approaches.

The variant c.11+5G>A was analysed  in several models: patients, human RPE cell line (ARPE19), human telomerase  immortalized RPE cell line (hTERT-RPE1), photoreceptor precursor cells  derived by   iPSC differentiation as well as iPSC from patients induced to fofferentiate into RPE cells.

The performed analysis included RPE markers gene and protein levels, Immunochemistry assays, analysis of proteomic data, PACBIO long-read genome sequencing and analysis and Statistical evaluations.

The most relevant results evidenced that the c.11+5G>A variant of RPE65 shows an exon elongation in vitro; its splicing defect is less noticeable in patient-derived iPSCs  and photoreceptor precursor cells, while reduces mRNA expression resulting in RPE65 absence, Moreover gene expression analysis of RPE specific markers by RT-  PCR   or qPCR , and   RPE specific proteins by ICC  or Western blot ( did not show differences between control and patient cells. Instead,  the authors referred relevant difference  between both RPE lines, regarding thee intensity of the cell pigmentation, and the morphology of  patient-derived RPE cells  seemed 1.5 times   bigger   than control-derived RPE cells. The analysis by Long-read genome sequencing does not revealed  other causative variant in the patient-derived  DNA. This analysis did not  identify any other possible genetic cause for the reduction in RPE65 expression. 

Moreover different luciferase reporter constructs were performed to   confirm the effect of the c.11+5G>A variant in gene expression, and the obtained results  suggest that the c.11+5G>A variant-mediated mRNA expression reduction is cell-context dependent.

Taking into account this amount of results and the very thorough approach the article could be a starting point to better ivestigate the pathways by which several variants of such disease lead to an impairment of retinal functions.

Reviewer 2 Report

The manuscript « The predicted splicing variant c.11+5G>A in RPE65 leads to a reduction of mRNA expression in a cell-specific manner » by Vasquez-Dominguez et al.  report explanation of the pathomechanism responsible for degeneration of the retina in LCA patients carried the c.11+5G>A variant.

 Here the paper do not question the pathogenicity of the c.11+5G>A but the pathomechanism of this variant. They demonstrated that c.11+5G>A was responsible for a 124nt exon elongation by midigene splicing assay and in the patient derived iPSCs and photoreceptor precursor cells (PPCs). The mutant transcript is barely detectable. The level of expression was too low to explain the pathogenicity.

They evidenced that the splicing defects can occur in a tissue and/or cell-type manner, concept now accepted and accurated. They contribute to highlight this point.

In P3-RPE It is more difficult or impossible to see this aberrant transcript even with cycloheximide treatment. To explain the disease, they argue that it is not the production of this aberrant transcript which is responsible for the disease but the low quantity of RPE65 transcript and by consequence the absence of protein.

In the discussion they add interesting points concerning regulatory elements and the possibility that the variant will be localized in regulatory element to either disrupt an enhancer or activate a silencer.  

The manuscript is clear, the figures are appropriates and easy to interpret.

The quantity and quality of this work is high level. All the experiments are precise and answer one by one the questions asked.

Minor comments:

 There is no discussion about the presence of the WT RPE65 transcript in all condition in iPSC, PPC and P3-RPE even in Patient with an homozygote variant.  

In Figure 5: Why no MERTK appear in western blot with Bovine control? You gave an explanation for Histone H3 but not for MERTK.
